# Bias Analysis in Unconditional Image Generative Models

## Abstract

The widespread usage of generative AI models raises concerns regarding fairness and potential discriminatory outcomes. In this work, we define the bias of an attribute (e.g., gender or race) as the difference between the probability of its presence in the observed distribution and its expected proportion in an ideal reference distribution. Despite efforts to study social biases in these models, the origin of biases in generation remains unclear. Many components in generative AI models may contribute to biases. This study focuses on the inductive bias of unconditional generative models, one of the core components, in image generation tasks. We propose a standardized bias evaluation framework to study bias shift between training and generated data distributions. We train unconditional image generative models on the training set and generate images unconditionally. To obtain attribute labels for generated images, we train a classifier using ground truth labels. We compare the bias of given attributes between generation and data distribution using classifier-predicted labels. This absolute difference is named bias shift. Our experiments reveal that biases are indeed shifted in image generative models. Different attributes exhibit varying bias shifts' sensitivity towards distribution shifts. We propose a taxonomy categorizing attributes as *subjective* (high sensitivity) or *non-subjective* (low sensitivity), based on whether the classifier's decision boundary falls within a high-density region. We demonstrate an inconsistency between conventional image generation metrics and observed bias shifts. Finally, we compare diffusion models of different sizes with Generative Adversarial Networks (GANs), highlighting the superiority of diffusion models in terms of reduced bias shifts.

## 1 Introduction

Generative AI models have achieved realistic generation qualities for various modalities including text (Touvron et al., 2023; OpenAI, 2023), image (Ramesh et al., 2022; Rombach et al., 2022; Esser et al., 2024), audio (Kreuk et al., 2023), and video (Ho et al., 2022; Singer et al., 2023). They are consequently employed for commercial uses and are available to every internet user across the world. The widespread use of these high-performing models, along with the potential social biases embedded in their generation, increase the risk of discriminatory outcomes. Taking image generation as an example, Growcoot (2023) and Tiku et al. (2023) report racial and gender biases in popular text-to-image (T2I) systems including DALL-E (Ramesh et al., 2021), Stable Diffusion (Rombach et al., 2022) and Midjourney (https://www.midjourney.com).

We define the bias of an attribute (e.g., gender or race) as the difference between the probability of its presence in the observed distribution and its expected proportion in an ideal reference distribution. The ideal reference distribution may be based on social norms or population statistics, etc. A widely studied problem is gender or racial bias with respect to occupations (Cho et al., 2023; Bianchi et al., 2023; Luccioni et al., 2023; Friedrich et al., 2024). Depending on the context, previous works use equality or U.S. labor statistics as the ideal reference distribution. In these cases, the typical analysis protocol consists of generating facial images based on text prompts containing occupation information, using pre-trained models to assign gender or racial labels for the generated images, followed by measuring and assessing the degree of biases in the images given a certain occupation.

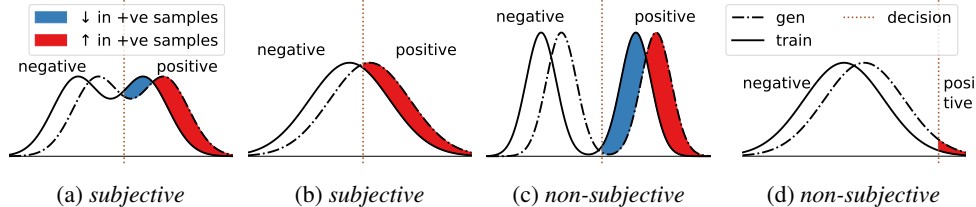

(a) *subjective*      (b) *subjective*      (c) *non-subjective*      (d) *non-subjective*

Figure 1: **Illustrations depicting bias shift.** The plots represent the distributions of samples with respect to the likelihood of an attribute (solid for training data, dashed for generation). The decision boundary (brown) binarizes the likelihood into positive and negative classes. In each subfigure, the generation distribution is translated from the training. Bias shift is the difference between red and blue areas. When the boundary falls in a low-density region (Figs. 1c and 1d), the bias shifts tend to be small, and vice versa (Figs. 1a and 1b). Detailed discussion is in Section 4.4 with distributions obtained from real datasets.

Other studies have compared social biases between generated images and training datasets of generative AI models, with mixed findings. Friedrich et al. (2024) report that images generated by Stable Diffusion (Rombach et al., 2022) show cases of bias and even bias amplification compared to the training data (LAION-5B) (Schuhmann et al., 2022). On the other hand, Seshadri et al. (2023) conduct similar experiments and discover that bias shift can be mainly attributed to discrepancies between training captions and model prompts.

Although analyzing biases empirically in publicly available generative AI models is of practical significance, identifying the origin of these biases remains a challenge. Modern generative AI systems are complex and generative biases can stem from various sources, such as biased datasets (Schuhmann et al., 2022; Karkkainen & Joo, 2021), the conditioning process (including textual prompts, and guidance (Dhariwal & Nichol, 2021; Ho & Salimans, 2022)), pre-trained modules (including CLIP (Radford et al., 2021) and VAE (Kingma & Welling, 2014)), and inductive bias of the generative models (e.g., diffusion process (Ho et al., 2020), generative adversarial training (Goodfellow et al., 2014)). While biases in pre-trained models (Bommasani et al., 2021; Alabdulmohsin et al., 2024) and datasets (Schuhmann et al., 2022) have been widely studied, the impact of inductive biases in generative models remains underexplored. Thus, in our experiments, we focus on *unconditional pixel-level* image generative models *without any guidance during training or inference*.

We propose a standardized evaluation framework that employs attribute classifiers to study bias shifts from training to generated data distributions in unconditional image generative models. Training the classifiers requires ground-truth labels for the training and validation sets; hence, our framework is applicable to any supervised learning dataset. We train unconditional image generative models using the training set and unconditionally generate images. We then use the trained classifiers to predict attribute labels for each generated image. We compare the bias for each attribute between the training and generated data distributions using classifier-predicted labels. We refer to this absolute difference as the *bias shift*. If bias shift is close to zero, there is no systematic bias exhibited in image generative models. We analyse the bias shifts on two real image datasets, CelebA (Liu et al., 2015) and DeepFashion (Liu et al., 2016).

Our findings reveal that bias shifts vary in magnitude across different attributes, indicating varying levels of sensitivity to distribution change between generation and training data. We categorize attributes as *subjective* (high sensitivity) and *non-subjective* (low sensitivity) sets, based on the relative sample density at the classifier's decision boundary. If the classifier is confident in its predictions — in other words, the decision boundary lies in a lower-density region (corresponding to *non-subjective* attributes), bias shifts tend to be smaller, and vice versa. Fig. 1 shows translation distribution shift as an example to introduce this idea.

Our bias analysis framework yields the following observations: 1) Biases of attributes shift between training and generation distributions for unconditional image generative models. The magnitude of bias shift is correlated with the *subjectivity* of the attribute. 2) Selecting the checkpoint based on image generation metrics considering quality, diversity, and novelty (FID (Heusel et al., 2017), KID (Binkowski et al., 2018), and FLD (Jiralerspong et al., 2023)) does not guarantee the smallest

bias shifts. Bias should be treated as an independent issue when evaluating generations. 3) BigGAN models (Brock et al., 2019) have larger bias shifts compared to diffusion models, despite having similar image generation metrics. 4) Bias shifts in smaller diffusion models tend to increase over extended training steps, even though image generation metrics remain relatively stable.

## 2 RELATED WORKS

**Bias in Image Generation**  Previous studies focus on social biases in image generation, often concluding that these models are unfair (Friedrich et al., 2024; Cho et al., 2023) or fail to reflect real-world biases as observed in U.S. labor statistics (Luccioni et al., 2023; Bianchi et al., 2023). A commonly examined bias is gender or race to occupation. However, different studies select various public models and develop their own evaluation benchmarks. For example, Cho et al. (2023) investigate minDALL-E (Kim et al., 2021), Karlo (Lee et al., 2022), and Stable Diffusion (Rombach et al., 2022) v1.4, while Luccioni et al. (2023) use Stable Diffusion v.1.4, v.2, and Dall-E 2 (Ramesh et al., 2022). However, there are different components in T2I models that may contribute to generative biases. We focus on the unconditional image generative model that directly outputs the generations, without text conditioning or guidance.

**Bias Shift between Train and Generation**  Few studies attempt to compare bias between the generation and training distributions. These efforts often rely on publicly available Stable Diffusion models, comparing generated outputs with the LAION-5B training set (Schuhmann et al., 2022), a large-scale dataset lacking explicit attribute labels. Given a text prompt, Friedrich et al. (2024) select a subset of LAION-5B based on pre-trained image-prompt similarity, then compare the bias between this subset and the images generated using the same prompt. In contrast, Seshadri et al. (2023) select subsets based on keywords in image captions, which may overlook relevant images. To avoid this large-scale dataset search and subset comparison, we train generative models using datasets with labeled attributes, ensuring reliable bias estimation across both the training and generation.

**Bias-related Attribute Label Prediction**  To calculate bias in generation, the generated images need to be assigned attribute labels, which is non-trivial in the case of unconditional generation. Some studies (Bianchi et al., 2023) infer the labels in the representation space of self-supervised learning models, for example, CLIP (Radford et al., 2021). Some methods use pre-trained vision language models and conduct zero-shot text generation. Cho et al. (2023) use BLIP-2 (Li et al., 2023) and get the label through visual question answering (VQA). Luccioni et al. (2023) use BLIP with VQA task and ViT (Dosovitskiy et al., 2021) with image captioning task. However, pre-trained models introduce their own biases (Bommasani et al., 2021; Alabdulmohsin et al., 2024), rendering the predicted labels unreliable for accurate bias evaluation. Some approaches (Friedrich et al., 2024) train an attribute classifier on other available supervised learning datasets. In our case, we train the classifier on the same dataset used for bias analysis, resulting in more accurate predictions.

## 3 BIAS EVALUATION METHOD

### 3.1 BIAS DEFINITION

In this work, bias for an attribute is defined as the difference between the probability of its presence in the observed distribution and its expected proportion in an ideal reference distribution.

Considering a set of binary attributes[1] $\mathcal{C}$ for which we want to study bias, each image in the dataset is annotated for every attribute. Given an attribute $C \in \mathcal{C}$, we consider the positive case $C = 1$ in the following. We can set an ideal probability $P^{\text{ideal}}(C = 1)$ for attribute $C$ as the reference probability, depending on the context. We denote the probability of this attribute in the data distribution as $P^{\text{data}}(C = 1)$. We can use either $P^{\text{train}}(C = 1)$ or $P^{\text{val}}(C = 1)$ as an estimation for $P^{\text{data}}(C = 1)$ and compare with the reference probability to determine degree of bias. For example, we define the bias of the data distribution relative to $P^{\text{ideal}}(C = 1)$ as

$$B^{\text{data}}(C = 1) = P^{\text{data}}(C = 1) - P^{\text{ideal}}(C = 1). \tag{1}$$

---

[1]The use of binary attributes can be extended to $K$-way attributes by binarizing the $K$-way attributes as $K$ 1-vs-all binary attributes.

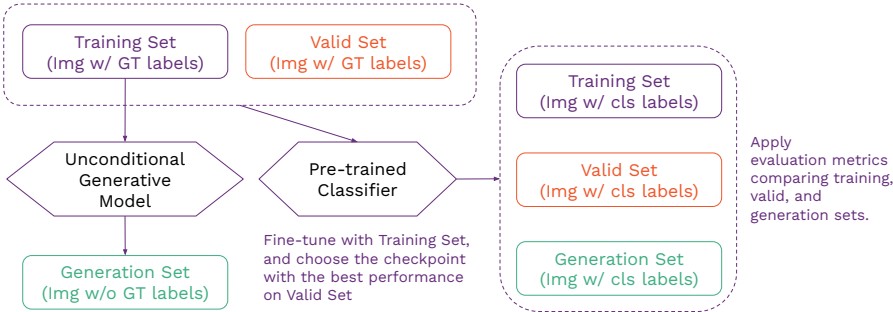

Figure 2: **Bias evaluation framework.** Unconditional generative models are trained on the training set. The pre-trained classifier is fine-tuned on the training set and validated on the validation set using ground truth labels and is then used to classify training, validation, and generation sets. The bias evaluation metrics are calculated based on the classifier-predicted labels.

To get the bias on the generation set, we need to calculate the proportion for this attribute in the generation set $P^{\text{gen}}(C = 1)$. We can then measure the bias in the generation

$$B^{\text{gen}}(C = 1) = P^{\text{gen}}(C = 1) - P^{\text{ideal}}(C = 1). \tag{2}$$

We also define the conditional bias. Given a binary anchor attribute $A \in \mathcal{C}$, the bias of attribute $C$ conditioned on $A = 1$ is the conditional probability $P(C = 1 \mid A = 1)$. Similarly, the conditional bias in the data distribution is $P^{\text{data}}(C = 1 \mid A = 1) - P^{\text{ideal}}(C = 1 \mid A = 1)$. The conditional bias in the generation distribution is $P^{\text{gen}}(C = 1 \mid A = 1) - P^{\text{ideal}}(C = 1 \mid A = 1)$.

## 3.2 BIAS EVALUATION FRAMEWORK

Fig. 2 illustrates our proposed bias evaluation framework. We train image generative models for unconditional image generation using only images from the training set, without feeding ground truth labels into the models. We generate 10,000 images for each checkpoint during training. To calculate the proportion for each attribute in the generation distribution, we require attribute labels for the generated images. We apply a trained classifier, developed using the training and validation sets with ground truth labels, to the generated images to obtain classifier-predicted attribute labels.

The trained classifier inevitably introduces errors, meaning the predicted labels may not match the ground truth labels for all images. To ensure consistent bias estimation across different sets, we use the trained classifier to predict attribute labels for training and validation sets. In addition, we use $P^{val}(C = 1)$ to estimate the probability of attribute $C$ in the data distribution, as the classifier may overfit to the training set. By adopting these techniques, we aim to minimize the potential bias introduced by the classifier in our bias evaluation framework for generative models.

Given a binary attribute $C \in \mathcal{C}$, we can therefore define **bias shift** between generation and training data as

$$B_{\text{shift}}(C = 1) = |B^{\text{gen}}(C = 1) - B^{\text{data}}(C = 1)| = |P^{\text{gen}}_{\text{cls}}(C = 1) - P^{\text{val}}_{\text{cls}}(C = 1)|. \tag{3}$$

The subscript cls stands for using classifier-predicted labels. In bias shift, the expected probability for positive attribute $C = 1$ in an ideal reference distribution $P^{\text{ideal}}(C = 1)$ is canceled out. Bias shift remains the same regardless which ideal bias reference we select. If bias shift is close to 0, then the generation distribution and the training distribution exhibit the same level of bias for the given attribute.

**Bias shift** evaluates changes in bias between data and generation distribution for each attribute considered in the study. To provide an overall understanding of the magnitude of bias shift across all attributes, we propose to use the average of bias shift across attributes. **Average bias shift (ABS)** evaluates the overall bias shift magnitude across all attributes considered between the training and the generated data distributions. This value represents the absolute difference between probabilities and is expressed as a percentage. We define this metric as

$$\text{ABS} = \mathbb{E}_{C \in \mathcal{C}} B_{\text{shift}}(C = 1). \tag{4}$$

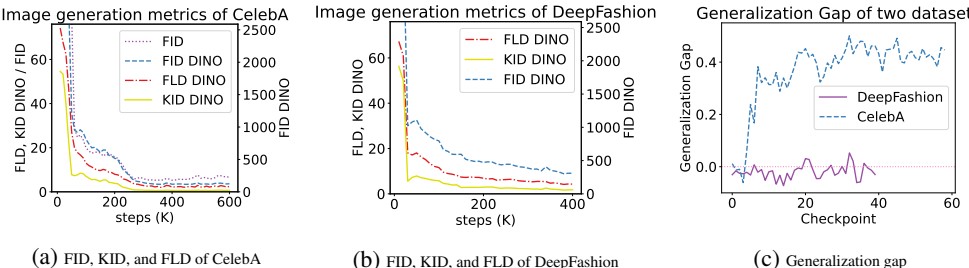

(a) FID, KID, and FLD of CelebA   (b) FID, KID, and FLD of DeepFashion   (c) Generalization gap

Figure 3: **Evaluation metrics for image generation throughout training.** In 3a and 3b, FID, KID, and FLD values converge to small values showing the good quality of generated images and good coverage of modes of the training distribution. In 3c, the positive or slightly negative generalization gaps indicate that the trained models do not have severe memorization issues.

## 4   EXPERIMENTS

### 4.1   EXPERIMENTAL SETUP

**Datasets**   We apply our proposed bias evaluation framework to two real datasets – CelebA (Liu et al., 2015) and DeepFashion (Liu et al., 2016). CelebA (Liu et al., 2015) is a large-scale dataset with 200,000 celebrity facial images, each labeled with 40 binary attributes. It covers a wide range of facial features, from details (e.g., earrings, pointy nose) to outlines (e.g., hair color, gender, age). DeepFashion (Liu et al., 2016) is a clothes dataset with over 800,000 diverse fashion images. We use a subset with 26 fine-grained attribute annotations to train the classifier. We then study the bias shift over these fine-grained attributes. For both datasets, we follow the training/validation/test set split from the official release. More details about these datasets are in Appendix A.

**Backbone models in the framework**   We follow the setup from Dhariwal & Nichol (2021) to train unconditional ablated diffusion models (ADMs)[2]. We train models of varying sizes by adjusting the number of channels in the U-Net (Ronneberger et al., 2015) bottleneck layer (32 for tiny, 64 for small, and 256 for large), with proportional changes in each layer. In the following sections, we report the results of the large diffusion model if the model is not otherwise specified. We generate 10,000 images per checkpoint using 100 inference steps across training. We use a ResNext50 (32x4d)[3] based image classifier (Xie et al., 2017). We add a linear layer on top as the classification head and fine-tune the last 6 layers of the ResNext50 model. For comparison with a GAN model, we train a BigGAN (Brock et al., 2019) model[4] using the recommended settings. Implementation details are in Appendix B.

**Evaluation metrics for Image Generation**   We use some common metrics, e.g., FID (Fréchet Inception Distance) (Heusel et al., 2017) and KID (Kernel Inception Distance) (Binkowski et al., 2018), to evaluate the generated images. We use FLD (Feature Likelihood Divergence) and generalization gap (Jiralerspong et al., 2023) as two additional metrics to gauge the memorization level of the generative models. FLD provides a comprehensive evaluation considering not only quality and diversity, but also novelty (i.e., difference from the training samples) of generated samples. Positive generalization gap shows no overfitting to the training set. We adopt the implementation[5] of Jiralerspong et al. (2023) and follow their suggestion of using DINOv2 (Oquab et al., 2024) as the feature extractor to calculate FID, KID, and FLD. We also use a conventional FID implementation[6] to give a comparable value of how well the trained models are.

---

[2] https://github.com/openai/guided-diffusion
[3] The pre-trained model is from torchvision.
[4] https://github.com/ajbrock/BigGAN-PyTorch
[5] https://github.com/marcojira/FLD
[6] https://github.com/mseitzer/pytorch-fid

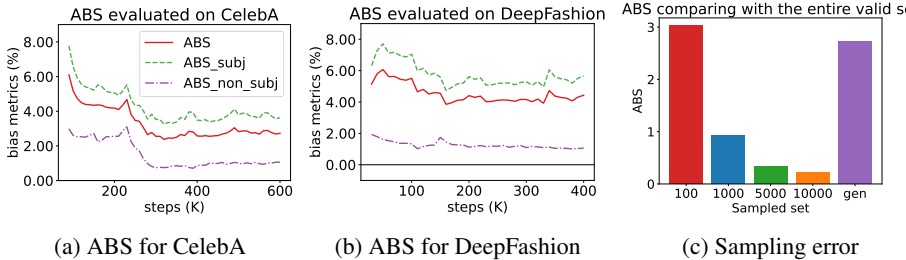

(a) ABS for CelebA     (b) ABS for DeepFashion     (c) Sampling error

Figure 4: **Average bias shift (ABS) for CelebA and DeepFashion.** For both datasets, shown in Figs. 4a and 4b, ABS over *subjective* attributes show a much larger bias shift than *non-subjective* ones. Fig. 4c presents that the error coming from sampling of 10K images is small enough, showing that the sampling randomness is not the only cause of bias shifts in generations.

Table 1: Attribute categorization of *subjective* and *non-subjective* for each dataset.

| Dataset | *subjective* attributes | *non-subjective* attributes |
|---------|-------------------------|------------------------------|
| CelebA | Rosy_Cheeks, Big_Nose, No_Beard, Narrow_Eyes, Arched_Eyebrows, High_Cheekbones, Bushy_Eyebrows, Black_Hair, Receding_Hairline, Brown_Hair, Straight_Hair, Bags_Under_Eyes, Pointy_Nose, Big_Lips, Mouth_Slightly_Open, Heavy_Makeup, Attractive, Smiling, Wearing_Lipstick, Wavy_Hair, Young, Oval_Face, | 5-o-Clock_Shadow, Bangs, Eyeglasses, Bald, Double_Chin, Wearing_Hat, Male, Blond_Hair, Gray_Hair, Mustache, Chubby, Pale_Skin, Sideburns,Goatee, |
| DeepFashion | Floral, Graphic, Embroidered, Solid, Long-sleeve, Short-sleeve, Sleeveless, Knit, Chiffon, Cotton, Maxi_length, Mini_length, No_dress, Crew_neckline,V_neckline, No_neckline, Loose, Tight, Conventional | Striped, Pleated, Leather, Faux, Square_neckline, Lattice, Denim, |

## 4.2 BACKBONE MODELS PERFORMANCE

**Diffusion Models** Figure 3 shows the image generation evaluation metrics for CelebA and Deep-Fashion datasets. In Figs. 3a and 3b, FID and KID converge to small values showing the good quality of generated images and good coverage of modes of the data distribution. FLD agrees with conventional metrics, showing no severe memorization issues in the generation. In Fig. 3c, the positive or slight negative values of generalization gap indicate that no overfitting is detected in the trained models. More discussions are in Appendix B.1.

**Classifier** For CelebA and DeepFashion datasets, the classification accuracy on the validation set for most attributes is over $80\%$. Overall, the average accuracy across attributes is $91.7\%$ for CelebA and $90.5\%$ for DeepFashion. Table 4 and Table 5 in Appendix B.2 show in detail the classifier performance for each attribute.

## 4.3 AVERAGE BIAS SHIFT EVALUATION

Fig. 4 presents the average bias shift (ABS) throughout training. The overall ABS is still perceivable when image generation metrics are small, indicating non-negligible bias shifts from the training to generation distributions. Looking closer into bias shift for each attribute (Figs. 8 and 9 in Section 4.6), we can categorize all attributes into two categories: *subjective* and *non-subjective*.

Taking CelebA as an example, intuitively, *non-subjective* attributes are those where classification judgements are consistent across populations, e.g., eyeglasses, wearing_hat, bangs, goatee, etc., while *subjective* ones are those where classification judgements differ significantly from one person to another, e.g., heavy_makeup, arched_eyebrows, attractive, oval_face, etc. We present the categorization of attributes in Table 1. In the following section 4.4, we will talk about the criteria for the attributes categorization.

Average bias shift (ABS) for *non-subjective* attributes (purple dashed lines in Fig. 4) converges to small values for both datasets, reaching $0.71\%$ for CelebA and $0.98\%$ for DeepFashion. However, *subjective* attributes exhibit significantly larger ABS, achieving minima of $3.25\%$ for CelebA and $4.73\%$ for DeepFashion.

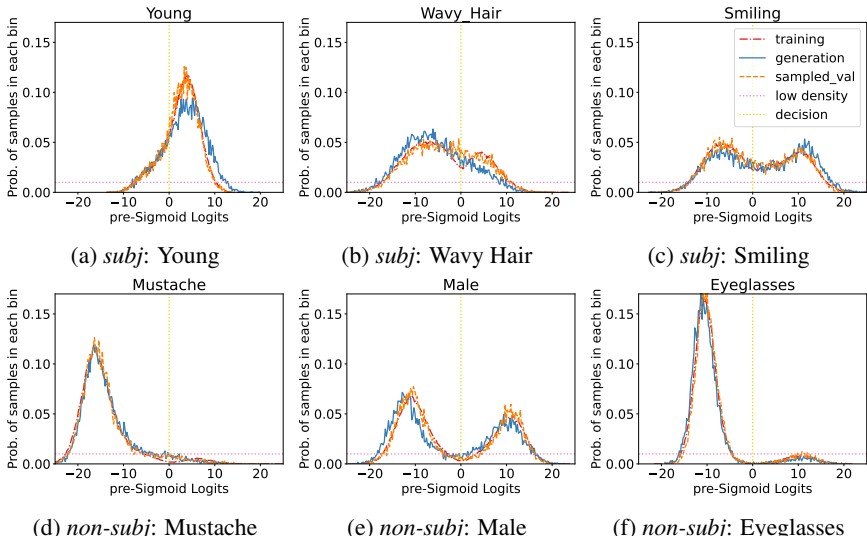

Figure 5: **CelebA classifier's pre-sigmoid logits distributions of selected *subjective* and *non-subjective* attributes.** The decision boundary for *subjective* attributes (Fig. 5a, 5b, and 5c) always falls in a high-density region, while for *non-subjective* attributes (Fig. 5d, 5e, and 5f) it falls in a low-density region.

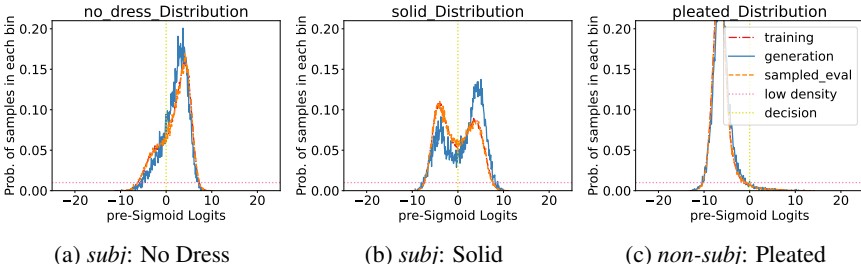

Figure 6: **DeepFashion classifier's pre-sigmoid logits distributions of selected *subjective* and *non-subjective* attribute.** The decision boundary for *subjective* attributes (Fig. 6a, 6b) always falls in a high-density region, while for *non-subjective* attribute (Fig. 6c) it falls in a low-density region.

Bias shifts do not consistently follow the image generation metrics, as illustrated by the comparison between Figs. 3 and 4. This misalignment highlights that models with superior image generation metrics are not necessarily less biased. Bias should be treated as an independent issue, distinct from quality and diversity. While diversity metrics typically assess the coverage of modes in the generated distribution, bias evaluation should focus on the relative proportions of these modes. For CelebA dataset, the bias evaluation metrics plateau between steps 110K and 210K, while the image generation metrics continue to improve. Similarly, for DeepFashion dataset, the image generation metrics continue improving during the whole training, while BSRs for both *subjective* and *non-subjective* attributes are stable with slight increases after about 200K steps.

To demonstrate that sampling 10,000 image generation is sufficient for a reliable statistical estimation, we present ABS between sampled subsets of the validation set and the full validation set on CelebA dataset in Fig. 4c. Additionally, we plot ABS between the generation set at the final checkpoint and the full validation set. The generation set has a much larger ABS compared to the sampled validation set with 10,000 images, emphasizing that the bias shifts observed in Figs. 4a and 4b exceed the variance introduced by the sampling process. This also suggests that using 10,000 images is sufficient to estimate bias shifts with minor errors.

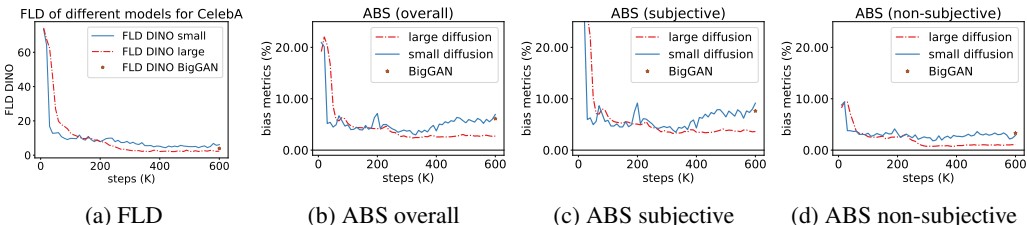

(a) FLD        (b) ABS overall        (c) ABS subjective        (d) ABS non-subjective

Figure 7: **FLD and ABS of different generative models on CelebA.** The small diffusion model has slightly worse image generation quality but much larger ABS for both *subjective* and *non-subjective* attributes compared to the large diffusion model. BigGAN has a similar FLD as the large diffusion model but has larger bias shifts.

### 4.4 BIAS SHIFTS' SENSITIVITY RELATES TO DECISION BOUNDARY

In this section, we analyze the classifier to explain why some attributes experience greater bias shifts than others, leading to the attribute taxonomy presented in Table 1.

Figs. 5 and 6 show the trained classifier's pre-sigmoid logits distribution for some attributes of CelebA and DeepFashion respectively. The distributions for all attributes are in Appendix B.2. These plots provide visualizations of how the data points are distributed in a projected uni-dimensional space. To estimate the empirical distributions, we use all the training images, 10,000 images sampled from the validation set, and all the 10,000 images in the generation set.

The main difference between *small bias shift* and *large bias shift* attributes is the density at the decision boundary. The distribution shifts for different attributes can manifest in various ways, but the decision boundaries for *large bias shift* attributes consistently fall in higher density regions compared to those for *small bias shift* ones. We thus use the density where the decision boundary falls in the validation distribution to categorize the attributes. Those with density more than 0.01 are categorized as *subjective*, and vice versa.

Bias shifts of *subjective* attributes are more sensitive to distribution shifts compared to *non-subjective* attributes. The distributions for *non-subjective* attributes still change between training and generation sets, but their effects on bias shifts are small. Since the decision boundary falls in a low-density region, it is more difficult to transport the density mass from one side of the boundary to the other. We find empirically that the distribution shifts between the training and generation distributions generally have low earth mover's distance (EM distance) (Rubner et al., 1998). Significant reweighting of well-separated modes would constitute a significant EM distance between training and generated distributions. For example, the distribution of `male` (Fig. 5e) shifts from training to generation, but the shifts are within each side of the decision boundary. This clear classification margin leads to small ABS for *non-subjective* attributes.

### 4.5 BIAS SHIFT IN DIFFERENT GENERATIVE MODELS

In this section, we compare the bias shifts for different sizes of diffusion models by changing the number of channels in the bottleneck layer of U-Net (32 for tiny, 64 for small, and 256 for large), and BigGAN model. Fig. 7 shows image generation metrics and bias evaluation metrics for different generative models. The tiny diffusion model cannot generate realistic images (check Appendix D for sampled images), making it unsuitable for our bias analysis framework. FLD for the small diffusion model is worse than the large diffusion model, while BigGAN achieves a similar FLD as the large diffusion model. However, ABS shows clear differences among generative models (See Figs. 7b, 7c and 7d).

Diffusion models have matched or even surpassed GAN models regarding image synthesis performance (Dhariwal & Nichol, 2021). We evaluate whether diffusion models also perform better than BigGAN regarding bias shifts. We observe that BigGAN exhibits a considerably larger ABS compared to the large diffusion model, despite having only slightly worse image generation performance according to FLD. This finding may be because the common understanding that GAN models suffer from mode collapse issues (Arjovsky et al., 2017).

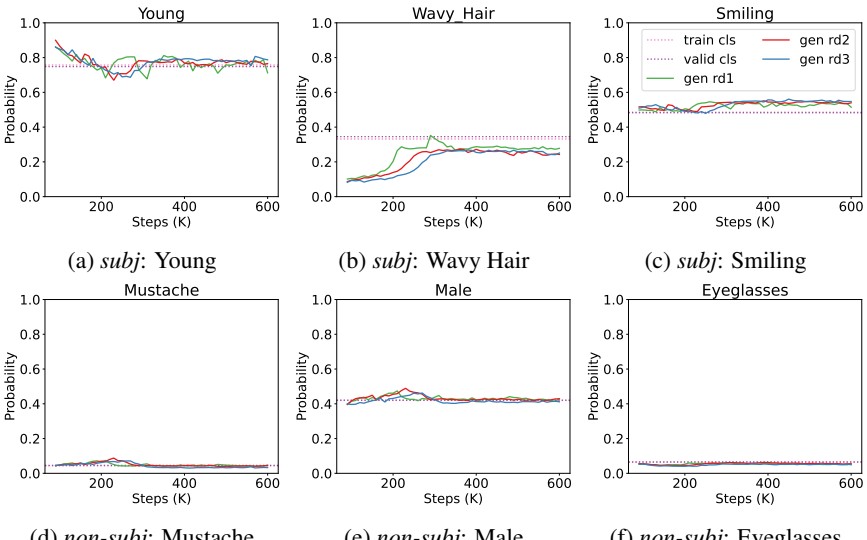

Figure 8: **Probabilities of selected *subjective* and *non-subjective* attributes for 3 different random seeds during training.** The probabilities of *subjective* attributes (Fig. 8a, 8b, and 8c) present a gap between generation and validation data while *non-subjective* ones (Fig. 8d, 8e, and 8f) do not.

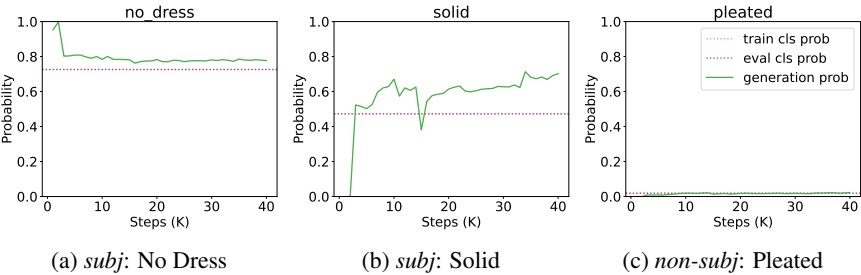

Figure 9: **Probabilities of selected *subjective* and *non-subjective* attribute during training.** The probabilities of *subjective* attributes (Fig. 9a, 9b) present a gap between generation and validation data while *non-subjective* one (Fig. 9c) does not.

We train different sizes of diffusion models to study the influence of the model size on bias shifts. The *small* diffusion model exhibits larger bias shifts compared to the *large* diffusion model. For the *small* diffusion model, we notice that the average bias shift increases after 300K steps although the FLD value does not change significantly. While BigGAN has better FLD, the bias shifts are similar to those of the *small* diffusion model at the end of the training. We also observe more fluctuations in bias evaluation metrics for the small diffusion model.

We present image samples generated from different models in Appendix D. The images generated by BigGAN and the small diffusion model are "more washed out" than those produced by the large diffusion model, showing fewer variations and less details.

## 4.6 ADDITIONAL RESULTS

**Per-attribute Bias Shift** Figs. 8 and 9 show probabilities for selected attributes in the generated data during training. Plots for other attributes are in Appendix C. Probabilities of *subjective* attributes generally exhibit values distinct from the classifier-predicted validation probabilities, resulting in bias shifts in Fig. 4.

*Subjective* attributes exhibit more fluctuations throughout training compared to *non-subjective* ones. While the probabilities for many attributes converge before 300K steps, young (Fig. 8a) still has fluctuations. A similar pattern is also witnessed in DeepFashion, where solid (Fig. 9b), as a

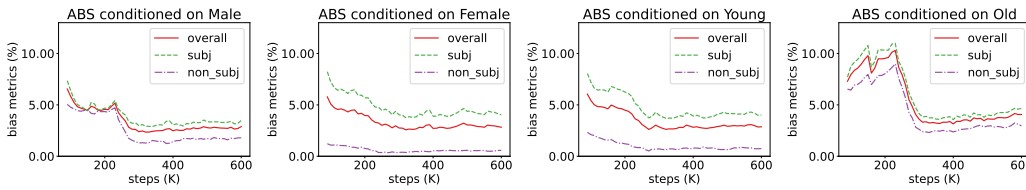

(a) conditioned on Male   (b) conditioned on Female   (c) conditioned on Young   (d) conditioned on Old

Figure 10: **ABS for conditional settings on CelebA.** Bias shifts conditioned on subjective attributes may exhibit different patterns as shown in Fig. 10d.

*subjective* attribute, also exhibits perceivable fluctuations. This suggests that extra caution is needed when handling certain *subjective* attributes using generative models.

We conduct several runs of training using different random seeds on CelebA dataset. There is randomness across different random seeds as the curves for each random seed vary. However, the probabilities of each attribute from distinct random seeds generally converge to the same value. Therefore, we report results for only one seed in other experiments.

**Bias Shift Evaluation Conditioned on Anchor Attributes**   Fig. 10 illustrates the conditional setting of bias shift evaluation. We focus on two demographic attributes, gender and age. According to our categorization proxy shown in Table 1, gender is *non-subjective*, while age is *subjective* in CelebA. This categorization may seem counterintuitive at first glance.

We acknowledge that it is not appropriate to naively binarize `gender` and `age`. However, due to the constraints of the era when the dataset was created, our analysis is restricted to binary `gender` and `age` attributes. By conducting an empirical analysis based on these binary attributes, we aim to highlight the importance of recognizing the fluidity of `gender` and the variability of `age`. It is important to note that the *subjective* and *non-subjective* categorization applies specifically to the image-label joint distribution presented in the CelebA dataset and is not universally applicable.

The bias change trends for probabilities conditioned on *non-subjective* attributes exhibit similarities to those of unconditioned probabilities (See Figs. 10a and 10b). However, we observe that the average bias shift for *non-subjective* attributes become larger when conditioning on `Old`, which is categorized as a *subjective* attribute in CelebA in our study. A possible explanation for this discrepancy is that the classifier-predicted labels of *subjective* attributes are not always accurate. Therefore, when conditioning on *subjective* attributes, classification errors propagate into the bias analysis pipeline, resulting in a distinct pattern of bias shifts.

## 5 CONCLUSION

This study focuses on bias shifts with regard to inductive biases of unconditional image generative models. We propose a standardized bias analysis framework applicable to any supervised learning dataset. Our experimental results show that different attributes have varying bias shifts in response to distribution changes. Attributes for which the classifier's decision boundary falls in a low-density area tend to have small bias shifts. We thus categorize all attributes into *subjective* and *non-subjective* sets. Our analysis results in the following observations: 1) Biases shift between training and generation distributions for unconditional image generative models. 2) Selecting the checkpoint with the best image generation metrics does not guarantee the smallest bias shifts. 3) BigGAN models and small diffusion models have larger bias shifts compared to large diffusion models, despite having similar image generation metric values.

We hope that our analysis for unconditional generative models can serve as a base framework allowing researchers to add other sources of bias such as conditioning (with ground-truth labels, text, etc.), guidance, pretrained modules, etc. in a gradual manner, and study their effects on bias shift in a systematic way.

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

# A DATASETS

CelebA (Liu et al., 2015) is a large-scale face attributes dataset with 200,000 celebrity images, each with 40 attribute annotations. The dataset includes 10,000 celebrities with 20 images for each. These attribute annotations cover a wide variety of facial characteristics, ranging from details (e.g., earrings, pointy noise, etc.) to outlines (e.g., hair color, gender, age, etc.). We list all 40 attributes in Table 2. Before feeding the training images to the model, we centre crop the images and re-size them to 128x128 pixels. Because of the crop, some attributes, e.g., Wearing_Necklace, Wearing_Necktie, are not visually grounded in the post-process images. Blurry is also an attribute that we do not include since we want the image generation quality to be good. We excluded these attributes in Table 1. We follow the Training/Validation/Test set split in the official release. Training set includes the images of the first eight thousand identities (with 160 thousand images). Validation set contains the images of another one thousand identities (with twenty thousand images). The remaining one thousand identities (with twenty thousand images) go for Test set. In our bias analysis framework, we only use the Training set and the Validadtion set.

DeepFashion (Liu et al., 2016) is a clothes dataset with over 800,000 diverse fashion images, including tops and bottoms. No footwears is in this dataset. Each image is associated with 1000 coarse attribute annotations about texture, fabric, shape, part, and style of the clothes. These attribute annotations are scrapped directly from meta-data of the images. They are thus very noisy and not reliable. Most of the attributes have less than 1% positive samples, making the classification problem very imbalanced. This dataset also provides a fine-grained annotation subset, where each image is associated with 26 find-grained attribute annotations. These attributes are presented in Table 2. We train a classifier on this subset and apply this trained classifier to the whole dataset and get classifier-predicted labels for each image. We follow the Training/Validation/Test set split in the official release. Unlike CelebA dataset, the split of DeepFashion dataset is random.

Table 2: Labeled attributes in CelebA and DeepFashion datasets. CelebA has 40 attributes and DeepFashion has 26 attributes.

| Dataset | Attributes |
| --- | --- |
| CelebA | 5_o_Clock_Shadow, Arched_Eyebrows, Attractive, Bags_Under_Eyes, Bald, Bangs, Big_Lips, Big_Nose, Black_Hair, Blond_Hair, Blurry, Brown_Hair,Bushy_Eyebrows, Chubby, Double_Chin, Eyeglasses, Goatee, Gray_Hair, Heavy_Makeup High_Cheekbones, Male, Mouth_Slightly_Open, Mustache, Narrow_Eyes, No_Beard, Oval_Face, Pale_Skin, Pointy_Nose, Receding_Hairline, Rosy_Cheeks, Sideburns, Smiling, Straight_Hair, Wavy_Hair, Wearing_Earrings, Wearing_Hat, Wearing_Lipstick, Wearing_Necklace, Wearing_Necktie, Young |
| DeepFashion | floral, graphic, striped, embroidered, pleated, solid, lattice, long_sleeve, short_sleeve, sleeveless maxi_length, mini_length, no_dress, crew_neckline, v_neckline, square_neckline, no_neckline, denim, chiffon, cotton, leather, faux, knit, tight, loose, conventional |

# B TRAINING DETAILS

## B.1 DIFFUSION MODELS

We follow the training setting of Dhariwal & Nichol (2021) to train the ablated diffusion models (ADMs). Hyperparameters and architecture selections are in Table 3. We train the diffusion using NVIDIA A100 40GB. The batch size per GPU is set to 16, and we use 8 GPUs to train. During training, we save checkpoint for EMA models every 10K steps. We use half precision (FP16) for training and inference.

For each saved checkpoint, we employ 100 steps in inference to generate 10K images from the Gaussian noise. We compare the two inference methods used in ADM (Dhariwal & Nichol, 2021),

Table 3: Hyperparameters and architecture selection for diffusion models

| lr | bsz | channel | res_block | dropout | diffusion_step | inference_step |
|------|-----|---------|-----------|---------|----------------|----------------|
| 1e-4 | 128 | 256 | 2 | 0.3 | 1000 | ddim100 |

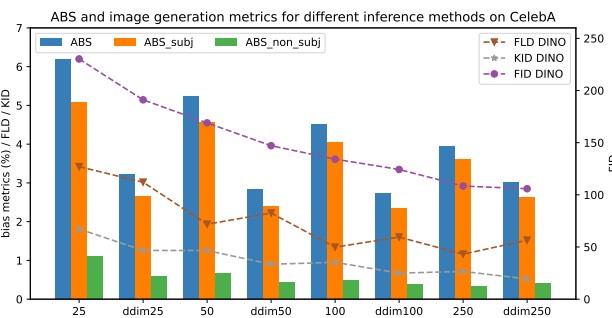

Figure 11: ABS and image generation metrics using different inference methods and inference steps on CelebA dataset. Images generated by DDIM have less bias shifts compared to those by Improved Diffusion Sampler. FID and KID also show the superiority of DDIM sampler.

one proposed by improved diffusion model (Nichol & Dhariwal, 2021), and DDIM (Song et al., 2021). The results on CelebA dataset are in Fig. 11. Images generated by the improved diffusion sampler exhibit more bias shifts than those from DDIM. Although FLD shows a slight improvement on improved diffusion sampler, DDIM works better in terms of FID and KID using the same steps of inference. Since we want to test less biased generations, we use DDIM with 100 steps during inference in our experiments.

In Fig. 3c, generalization gaps for CelebA and DeepFashion datasets are different. This is because the split of the dataset is in different ways. In CelebA dataset, the training and validation sets contain the faces of distinct sets of celebrities. In DeepFashion dataset, the training and validation samples are split randomly. The distribution difference between training and validation sets of CelebA is larger than that of DeepFashion.

## B.2 RESNET CLASSIFIERS

We employ a pre-trained ResNeXt model as the base model. We add a linear layer to top as the classification layer. We then fine-tune the last 6 layers of the pre-trained model as well as the classification layer using CelebA and DeepFashion dataset. We use AdamW optimizer and learning rate at 0.001. We follow a standard training procedure for the classifier training. We train the classifier on the train set (with ground truth labels) and choose the best classifier according to the average performance across all the considered attributes on the valid set (with ground truth labels). We use data augmentations to make the classifier more robust. The data augmentations include random horizontal flip, scaling and resizing, etc. This can help the classifier become more reliable when applied to the generation set. Previous work indicates that classifiers can amplify the discriminative biases in the training set (Zhao et al., 2017; Hall et al., 2022). We use the positive and negative sample ratio to reweigh the cross entropy loss terms. This acts as an upsampling of the minority samples and alleviates the label imbalance issue. We don't see the discriminative biases being amplified for most attributes according to Figs. 15 and 14 comparing the training ground truth probability and the validation classifier-predicted probability. The classifiers' performances for each attribute are listed in Tables 4 and 5. For both dataset, the accuracy for most attributes is over 80%. Figs. 12 and 13 show the pre-sigmoid logits distributions for each attribute in CelebA and DeepFashion datasets respectively.

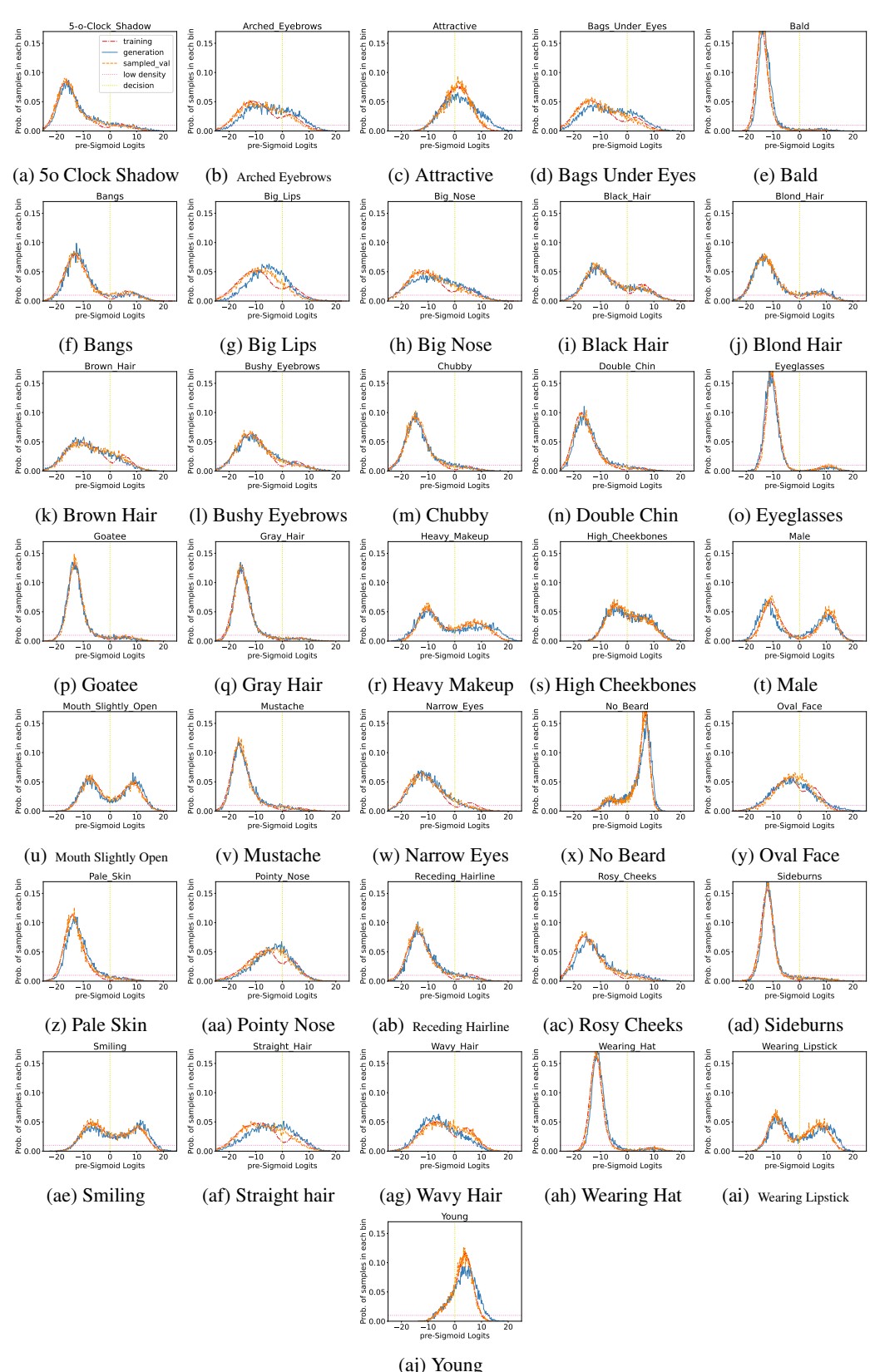

Figure 12: The pre-sigmoid logits distribution of each attribute in CelebA.

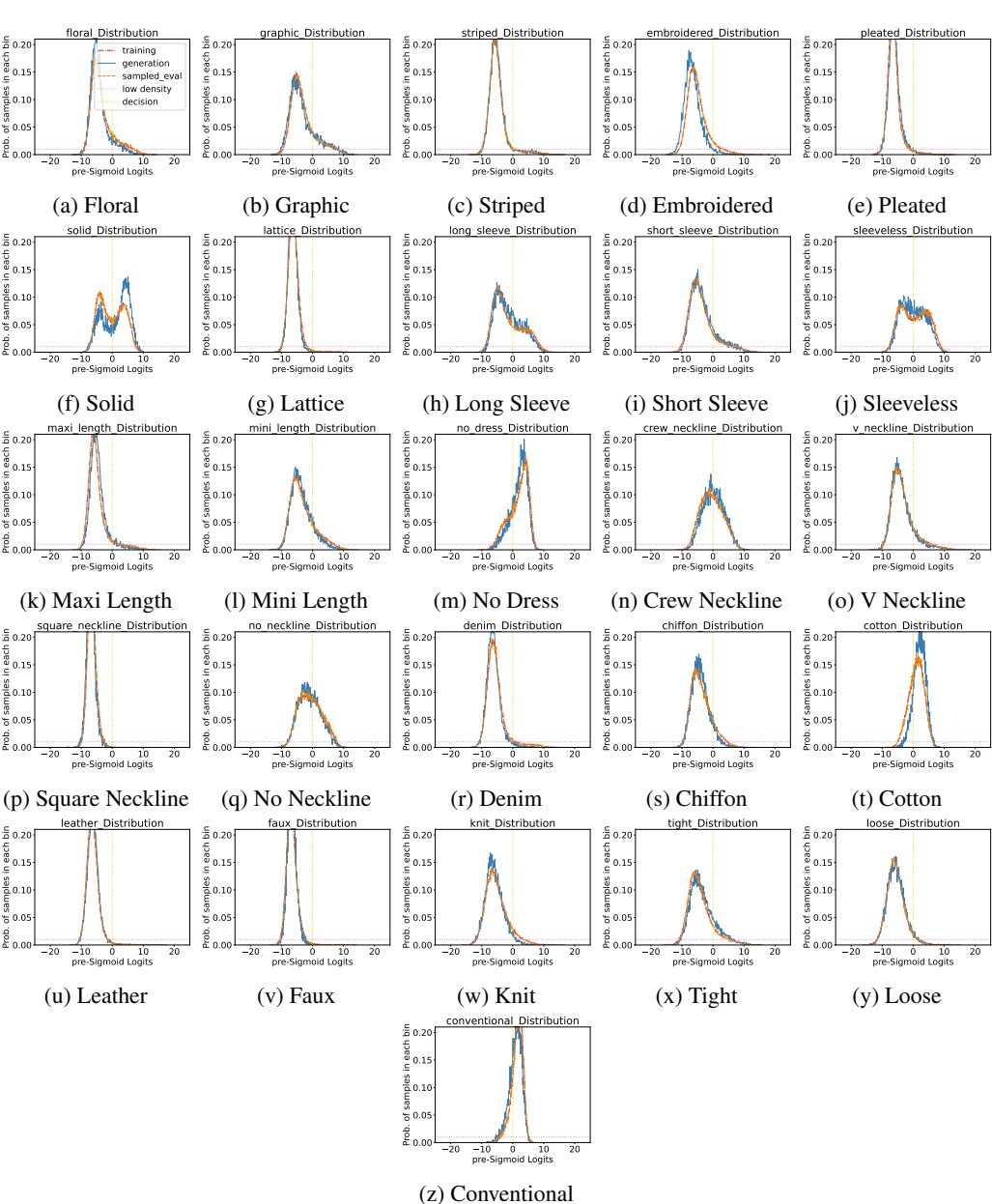

Figure 13: The pre-sigmoid logits distribution of each attribute in DeepFashion.

Table 4: Classifier performance on validation set of CelebA.

| Attr | Accuracy | Precision | Recall | F1 | AUPR |
|------|----------|-----------|--------|-----|------|
| Eyeglasses | 99.58 | 97.10 | 96.82 | 96.96 | 94.23 |
| Wearing_Hat | 98.98 | 86.31 | 93.19 | 89.62 | 80.75 |
| Bald | 98.92 | 73.33 | 74.94 | 74.13 | 55.47 |
| Male | 98.64 | 98.47 | 98.32 | 98.40 | 97.53 |
| Gray_Hair | 97.74 | 78.09 | 74.46 | 76.23 | 59.39 |
| Sideburns | 97.12 | 82.88 | 73.30 | 77.80 | 62.59 |
| Goatee | 96.61 | 76.83 | 77.25 | 77.04 | 61.03 |
| Double_Chin | 96.51 | 69.99 | 50.46 | 58.64 | 37.75 |
| Pale_Skin | 96.41 | 60.32 | 48.83 | 53.97 | 31.66 |
| Mustache | 95.90 | 60.78 | 53.14 | 56.70 | 34.66 |
| Blurry | 95.86 | 55.59 | 62.45 | 58.82 | 36.49 |
| Wearing_Necktie | 95.66 | 71.41 | 67.15 | 69.21 | 50.34 |
| No_Beard | 95.49 | 97.87 | 96.62 | 97.24 | 97.34 |
| Chubby | 95.35 | 65.18 | 51.73 | 57.68 | 36.67 |
| Bangs | 95.26 | 82.86 | 85.39 | 84.10 | 72.89 |
| Blond_Hair | 95.07 | 82.75 | 85.86 | 84.28 | 73.23 |
| Rosy_Cheeks | 94.64 | 64.32 | 48.45 | 55.27 | 34.69 |
| Receding_Hairline | 94.15 | 59.84 | 56.82 | 58.29 | 37.11 |
| 5-o-Clock_Shadow | 93.34 | 77.82 | 60.90 | 68.33 | 52.00 |
| Mouth_Slightly_Open | 92.83 | 92.97 | 92.07 | 92.52 | 89.42 |
| Wearing_Lipstick | 92.08 | 87.96 | 95.29 | 91.48 | 85.92 |
| Smiling | 91.50 | 90.73 | 91.80 | 91.26 | 87.25 |
| Bushy_Eyebrows | 91.42 | 72.05 | 65.03 | 68.36 | 51.84 |
| Heavy_Makeup | 91.19 | 86.20 | 92.17 | 89.08 | 82.50 |
| Narrow_Eyes | 90.97 | 42.41 | 56.57 | 48.48 | 27.25 |
| Wearing_Earings | 90.62 | 82.10 | 65.00 | 72.56 | 60.04 |
| Black_Hair | 89.60 | 71.52 | 83.33 | 76.97 | 63.07 |
| Wearing_Necklace | 86.98 | 43.51 | 26.71 | 33.10 | 20.46 |
| Young | 86.42 | 90.45 | 91.47 | 90.96 | 89.11 |
| High_Cheekbones | 86.09 | 83.47 | 86.10 | 84.76 | 78.11 |
| Brown_Hair | 83.41 | 66.70 | 62.42 | 64.49 | 50.70 |
| Bags_Under_Eyes | 83.33 | 64.93 | 42.73 | 51.54 | 39.63 |
| Arched_Eyebrows | 83.08 | 72.64 | 55.40 | 62.86 | 51.77 |
| Wavy_Hair | 83.06 | 66.23 | 79.04 | 72.07 | 58.15 |
| Straight_Hair | 81.97 | 56.09 | 56.70 | 56.39 | 40.71 |
| Big_Nose | 81.63 | 69.39 | 46.81 | 55.91 | 45.71 |
| Big_Lips | 81.28 | 37.00 | 31.57 | 34.07 | 22.17 |
| Attractive | 80.07 | 78.42 | 85.09 | 81.62 | 74.48 |
| Pointy_Nose | 72.97 | 52.86 | 47.24 | 49.89 | 40.00 |
| Oval_Face | 68.34 | 44.95 | 57.86 | 50.59 | 37.81 |

Table 5: Classifier performance on validation set of DeepFashion.

| Attr | Acc | Precision | Recall | F1 | AUPR |
|---|---|---|---|---|---|
| lattice | 99.48 | 100.00 | 50.00 | 66.67 | 50.52 |
| square_neckline | 98.97 | 0.00 | 0.00 | 0.00 | 1.03 |
| faux | 98.45 | 50.00 | 33.33 | 40.00 | 17.70 |
| leather | 97.94 | 0.00 | 0.00 | 0.00 | 1.03 |
| pleated | 97.42 | 40.00 | 50.00 | 44.45 | 21.03 |
| maxi_length | 96.91 | 96.00 | 82.76 | 88.89 | 82.03 |
| denim | 96.91 | 87.50 | 58.33 | 70.00 | 53.62 |
| striped | 96.39 | 55.56 | 62.50 | 58.82 | 36.27 |
| loose | 94.33 | 60.00 | 25.00 | 35.29 | 19.64 |
| knit | 92.27 | 52.63 | 62.50 | 57.14 | 35.99 |
| mini_length | 91.24 | 75.61 | 81.58 | 78.48 | 65.29 |
| graphic | 90.72 | 69.70 | 74.19 | 71.88 | 55.83 |
| embroidered | 90.72 | 36.36 | 26.67 | 30.77 | 15.37 |
| long_sleeve | 90.72 | 82.54 | 88.14 | 85.25 | 76.36 |
| short_sleeve | 90.21 | 66.67 | 73.33 | 69.84 | 53.01 |
| no_dress | 90.21 | 90.91 | 94.49 | 92.66 | 89.51 |
| solid | 88.14 | 88.89 | 88.00 | 88.44 | 88.41 |
| floral | 87.63 | 61.90 | 76.47 | 68.42 | 51.46 |
| tight | 87.63 | 61.29 | 61.29 | 61.29 | 43.75 |
| chiffon | 87.11 | 57.69 | 51.72 | 54.55 | 37.06 |
| v_neckline | 86.60 | 70.83 | 47.22 | 56.67 | 43.24 |
| sleeveless | 86.08 | 86.79 | 87.62 | 87.20 | 82.75 |
| conventional | 80.93 | 86.54 | 89.40 | 87.95 | 85.62 |
| no_neckline | 75.26 | 71.26 | 72.94 | 72.09 | 63.84 |
| cotton | 75.26 | 81.34 | 82.58 | 81.95 | 79.03 |
| crew_neckline | 71.65 | 59.30 | 71.83 | 64.97 | 52.91 |

## C  BIAS SHIFT ANALYSIS PER ATTRIBUTE

Figs. 15 and 14 show the bias probability for each attribute in CelebA and DeepFashion datasets respectively.

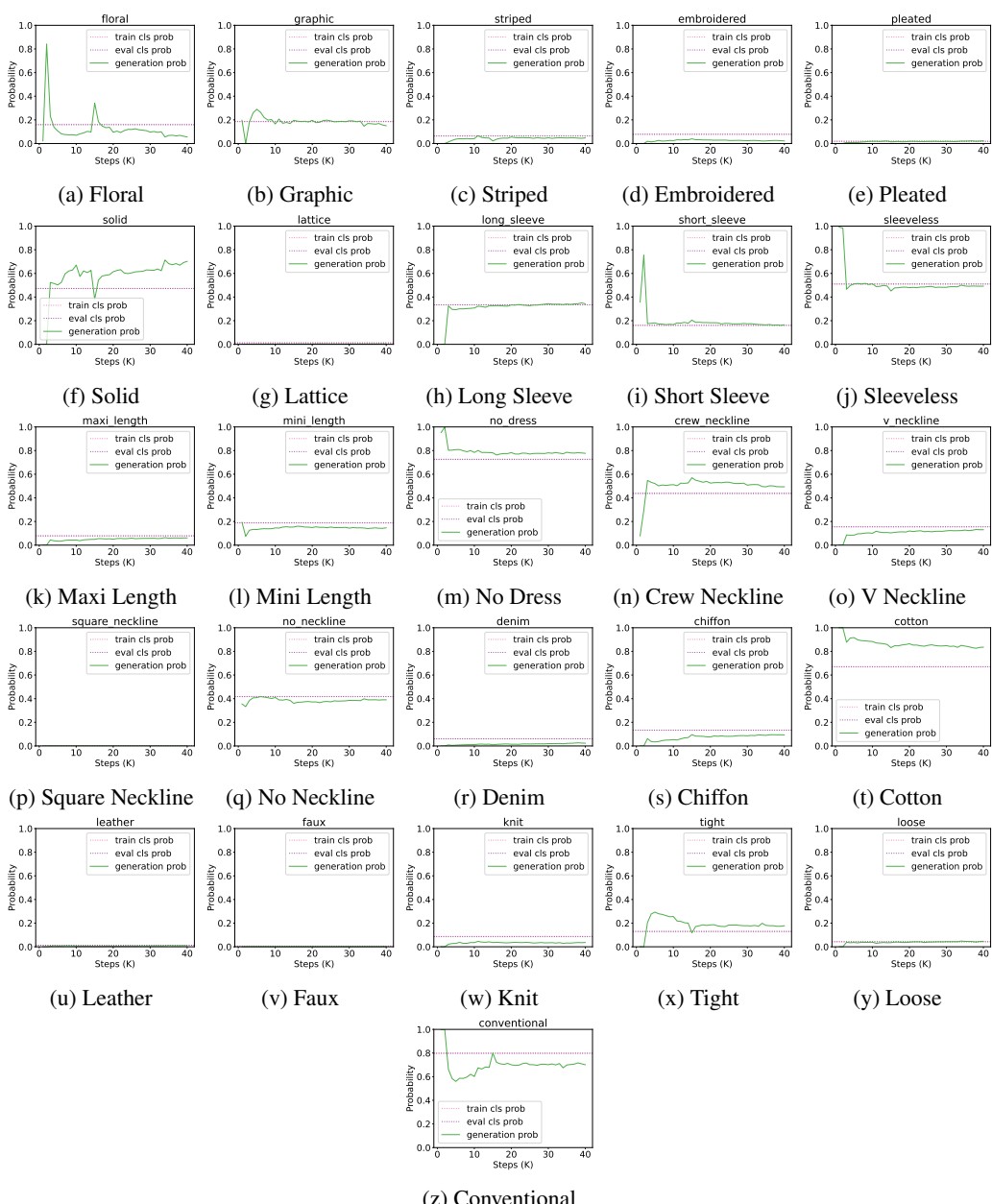

(a) Floral    (b) Graphic    (c) Striped    (d) Embroidered    (e) Pleated

(f) Solid    (g) Lattice    (h) Long Sleeve    (i) Short Sleeve    (j) Sleeveless

(k) Maxi Length    (l) Mini Length    (m) No Dress    (n) Crew Neckline    (o) V Neckline

(p) Square Neckline    (q) No Neckline    (r) Denim    (s) Chiffon    (t) Cotton

(u) Leather    (v) Faux    (w) Knit    (x) Tight    (y) Loose

(z) Conventional

Figure 14: Probabilities of attributes for DeepFashion dataset during training. Please note that it might seem like some of the subplots are missing the probability lines; they are actually very close to the x-axis, especially for *Square Neckline* and *Faux*.

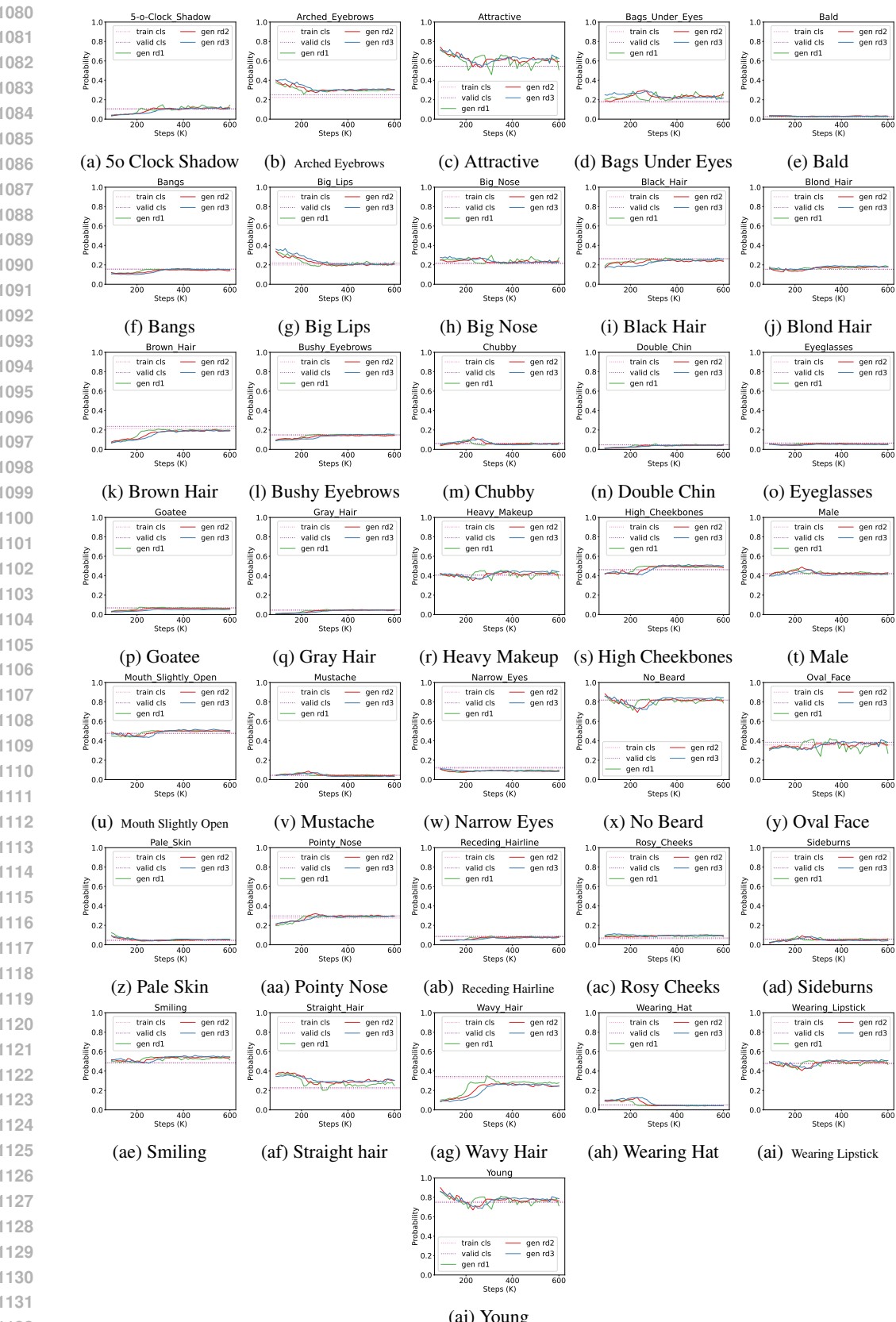

Figure 15: The probabilities of attributes in CelebA during training.

## D  SAMPLES OF GENERATED IMAGES

For different models and different dataset, we sample 80 images from the generation set and present them in Figs. 16, 17, 18, 19 and 20.

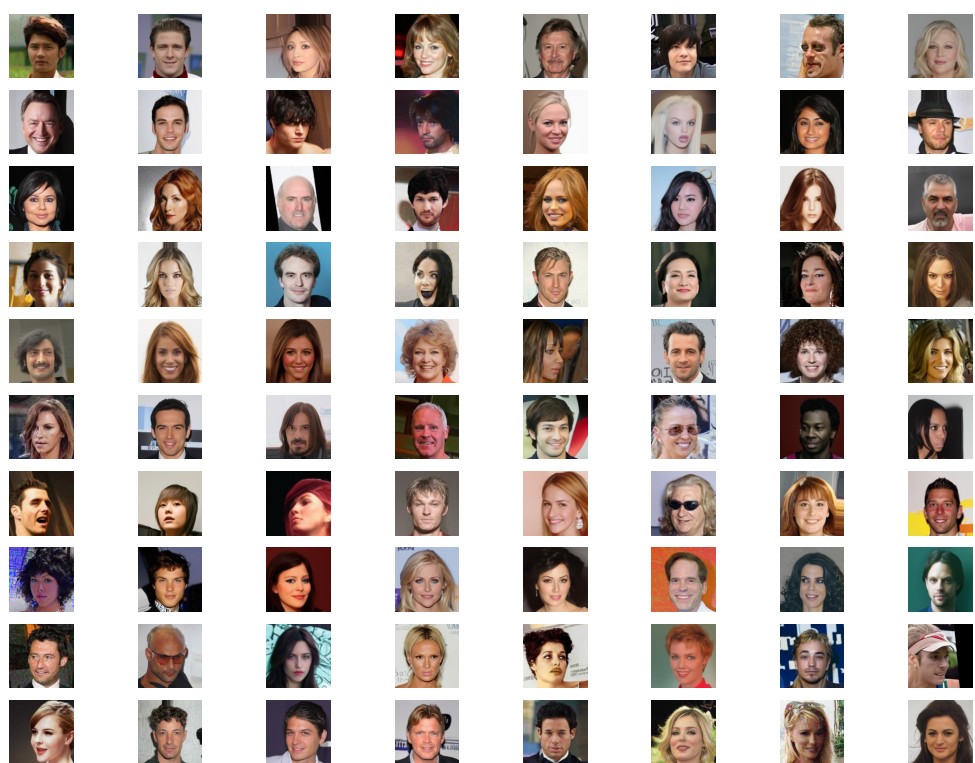

Figure 16: Image samples from large diffusion model generations on CelebA dataset.

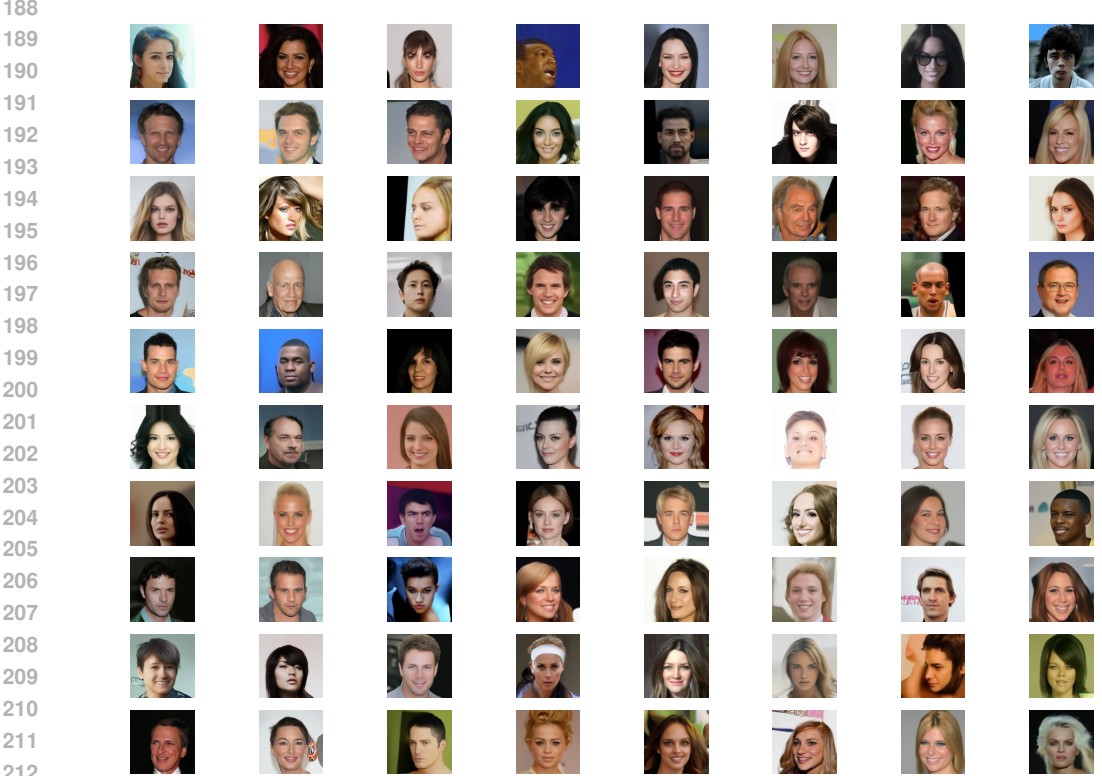

Figure 17: Image samples from the small diffusion model trained on CelebA dataset.

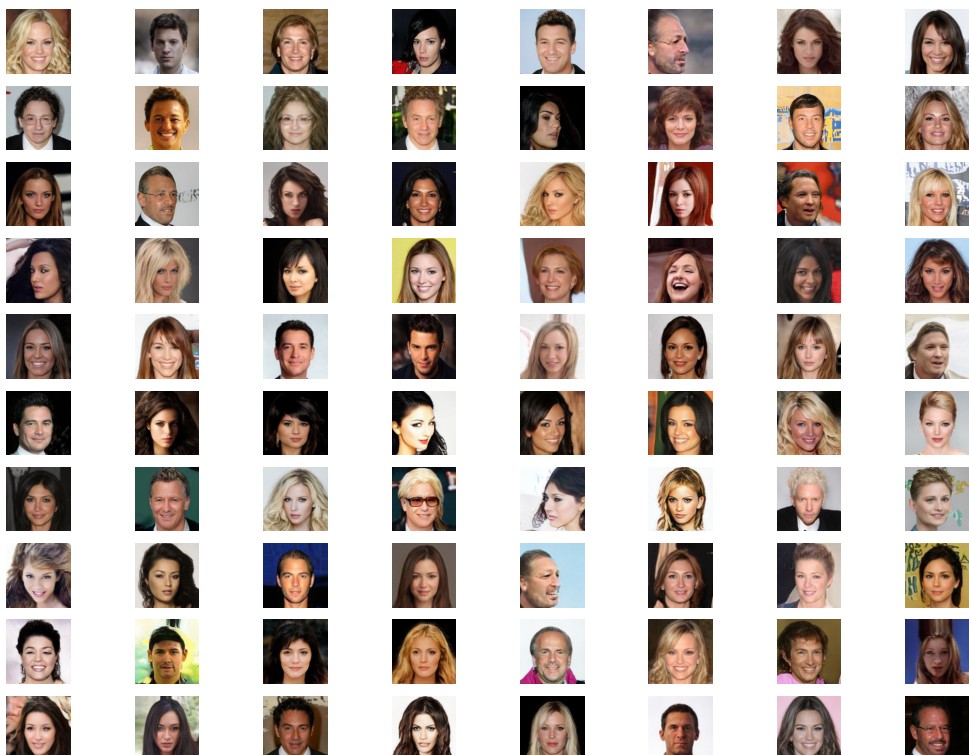

Figure 18: Image samples from the BigGAN model trained on CelebA dataset.

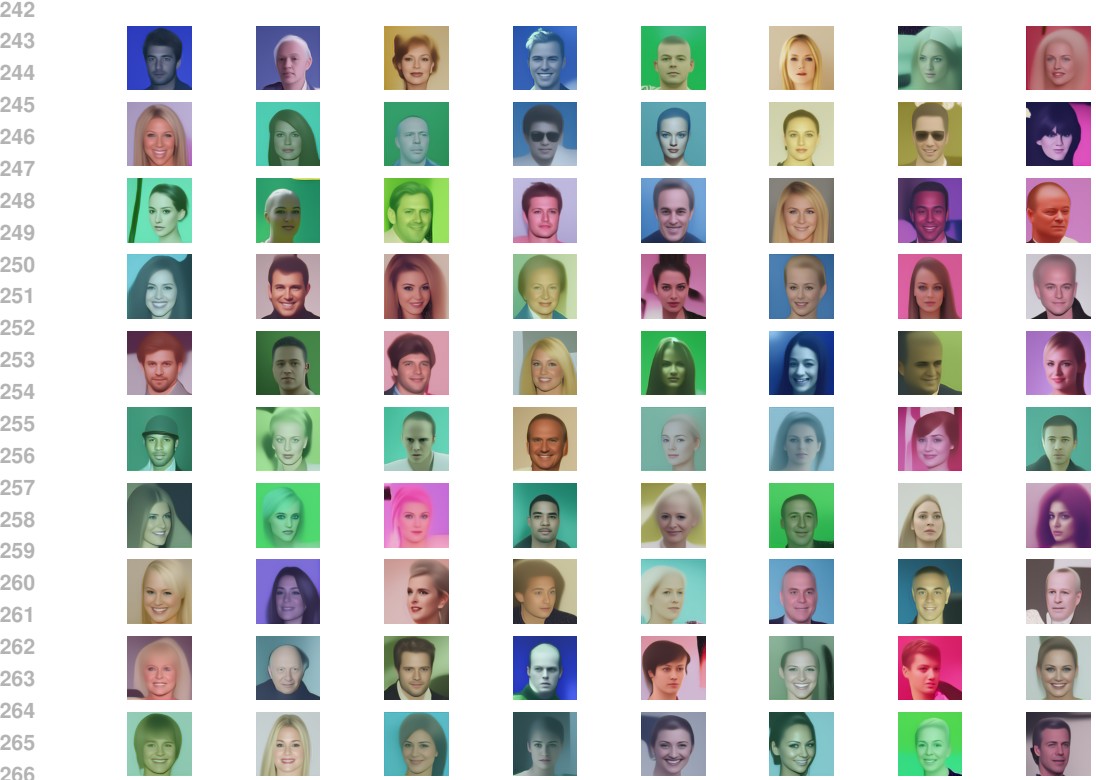

Figure 19: Image samples from the tiny diffusion model trained on CelebA dataset.

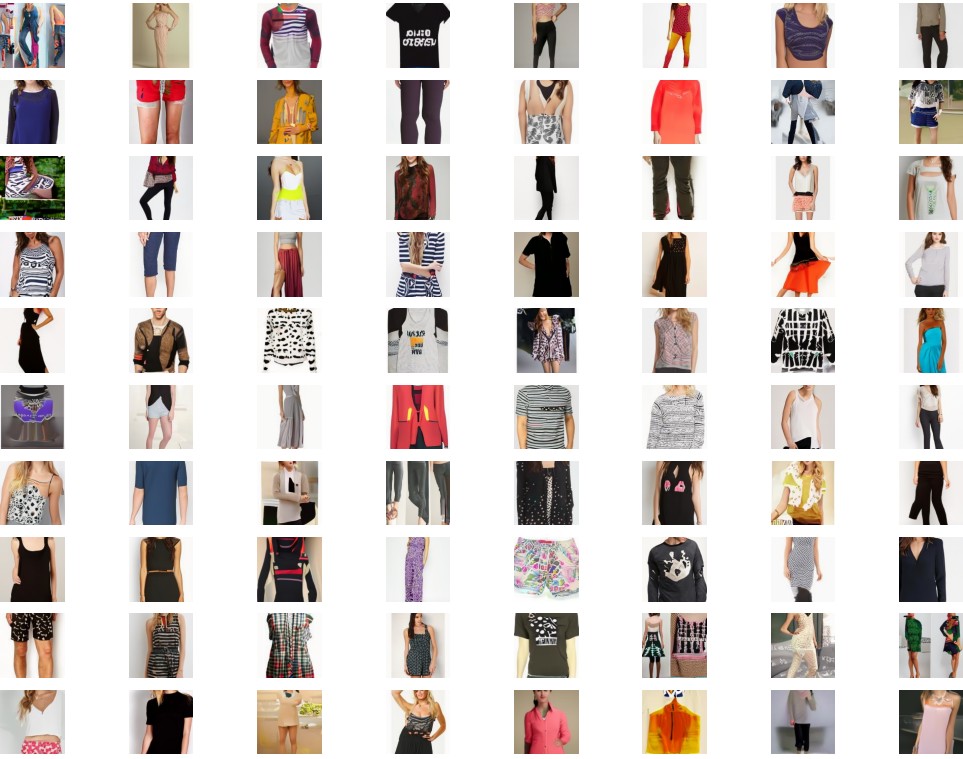

Figure 20: Image samples from the large diffusion model trained on DeepFashion dataset.

