# OpenReview forum: "Bias Analysis in Unconditional Image Generative Models"
_ICLR.cc/2025/Conference — Submitted to ICLR 2025_

### Official Review · Reviewer_QibE · 2024-11-02

**Soundness:** 2
**Presentation:** 2
**Contribution:** 1
**Rating:** 3
**Confidence:** 5

**Summary:**

This paper introduces an analysis of non-conditional generative models, GANs and Diffusions, for image generation. By using a classifier trained on the same dataset, biases are identified as subjective and non-subjective attributes.

**Strengths:**

The paper analyzes an important topic of bias in generative models. These models have been shown to learn the biases from their datasets, and this paper proposes a new angle towards understanding these biases.

The paper is generally well written and easy to follow.

The detailed analysis of logits seems to not have been studied before.

**Weaknesses:**

As a preliminary, I have reviewed this paper before for the NeurIPS SafeGEN workshop. I have reread this version of the paper, and my opinion has not changed.

Here's my concerns:
 - Calling attributes subject vs non-subjective is very strange. For example, "Pale skin" or "male" in CelebA being a non-subjective attribute is surprising. I would be convinced if there were a user study to validate these attributes are similarly subjective to humans, but as it stands I'm not convinced.
 - The raw bias shift is strikingly small. The subjective logits on figure 5c look extremely similar between the synthetic versus real data. Especially when comparing 5c to 5e, it's surprising that Male landed in non-subjective and Smiling did not.
 - Using the same dataset for training the generator and classifier is very problematic: it's self-contamination. The bias studied in this paper could come from: the dataset itself, the generator's training/architecture, or the classifier's training/architecture. Given the generator and classifier are mapped to the same data distribution, the inherit biases are muddled between the two. I would have liked to seen dataset splits where half the data is used to train the generator and half the classifier. That would improve the self contamination issue substantially.
 - Finally, the actual take-aways from the paper are fairly limited. Assuming my previous point were address, the fundamental why question is not answered: why are some attributes represented more/less in the synthetic distribution. It is somewhat useful to know that some attributes are, but I would be very interested to know how to predict which attributes would be over/under represented by just training a classfier.

**Questions:**

None

---

### Official Review · Reviewer_FbqA · 2024-11-03

**Soundness:** 2
**Presentation:** 3
**Contribution:** 2
**Rating:** 5
**Confidence:** 4

**Summary:**

This paper proposes an evaluation pipeline to analyze the bias shift of different generative models. The generative models are trained on the training dataset, and an attribute classifier is also pretrained on the same dataset. The attribute prediction difference between the original dataset and the generated images measures the bias shift. The paper also separates the attribute into two categories, subjective and non-subjective, to further analyze the insight of the bias shift.

**Strengths:**

In general, the analysis of the bias shift of different generative models is interesting, and the pipeline's high-level idea seems to be sound. The subjective/non-subjective study is also interesting. The paper includes a vast amount of empirical results for the analysis.

**Weaknesses:**

1. Some of the discussion in the method section (sec. 3) seems to be redundant or not tied to the paper. For example, what is the purpose of introducing $P^{ideal}$? Although it is canceled in the final equation, I don't think it is necessary to introduce such a term because, intuitively, $|{P^{gen} - P^{val}}|$ itself is sufficient to measure the bias shift. Introducing extra and probably unnecessary assumptions may overcomplicate the method and lead to confusion. Also, Section 3.1 also introduces a definition of "conditional bias," which is not discussed or studied in the rest of the paper. What is the purpose of introducing this definition?

2. The paper claims (L142) that "pre-trained models introduce their own biases, rendering the predicted labels unreliable for accurate bias evaluation." However, I disagree with this argument. I agree that the pre-trained model may be biased, but this reason does not invalidate them for performing attribute classification. Such a classifier serves as the expert in labeling attributes so that the most important criterion, if not the only criterion, should be classification accuracy. If any pre-trained models have outstanding attribute classification performance on the training/val dataset, I don't see why they shouldn't be used. Further, those pre-trained models can be finetuned on the training dataset (which this paper did) for an even better classification performance on specific datasets (e.g., CelebA), which can only benefit the bias shift analysis.

3. Further, the accuracy of the classifier is not sufficient for the analysis. Although the accuracies of the majority of attributes are 90%+, there is still a considerable amount of attributes on which the classifier performs unsatisfying. This fact is critical to the analysis, considering the listed subjective attribute examples are placed in the lower portion of the performance list. Lower accuracy may suggest higher analysis noise and larger ABS measuring error. Since the ABS for non-subjective attributes and subjective attributes are ~1% and 3-5%, the classifier with 91.7% (lowest attribute: 68.34%) or 90.5% (lowest attribute: 71.65%) accuracy is not good enough.

4. Further, the classifier is trained on the training set and directly applied to both training, valid, and generation sets. However, unlike training and valid sets are sampled from the same distribution, the generation set may have a different distribution than the original dataset. Thus, the classifier may suffer from distribution shift and/or visual domain generalization challenges, so the classifier may not be reliable on the generation set. This issue can further weaken the paper's analysis and conclusion.

5. Although the attempt to split the attribute into subjective and non-subjective groups is interesting, I am not convinced that the splitting method used in the paper (decision boundary-based) is valid. The decision boundary is closely connected with classifier accuracy, which can be further connected with analysis noise and measuring errors. Thus, those attributes with unclear boundaries are more likely to have higher ABS errors. Additionally, this splitting may not match human's definition of "subjective." Those "subjective attributes" to human definition (e.g., wearing glasses) may be easier to be classified so that they may have clearer boundaries. However, there is no guarantee of this, and the paper also does not have a complete list of subjective and non-subjective attributes to verify.

**Questions:**

Please refer to the Weaknesses.

---

> ### Comment · Reviewer_FbqA · 2024-11-27
>
> Many thanks to the authors for their detailed responses. The response addresses some of my concerns, and I have increased my rating. However, I don't think the response on the classifier selection is very convincing. On many attributes, the classifier can have high accuracy and align with previous literature, but this does not necessarily suggest that the analysis on those attributes with low accuracy is correct. Also, even though the generated data seems to have a similar distribution to the real data, it does not guarantee that the classifier trained on the real data will maintain its performance on the generated data. Literature (e.g., works in adversarial attacks) has shown that even a slight change in the data distribution may largely affect the model's performance. Thus, I suggest the paper perform a more detailed analysis of this.

---

> > ### Author Response · Authors · 2024-12-03
> >
> > Thank you for the feedback and acknowledgment of our justifications. We would greatly appreciate if the reviewer could provide further clarifications for the following:
> > 1. What specific type of detailed analysis would, in the reviewer's view, ensure that a classifier trained on real data maintains its performance when applied to generated data, particularly in the context of unconditional image generation?
> > 2. Is there any additional input from you to help refine the analysis on attributes with lower accuracy, beyond the methods we have already explored in the paper, particularly with respect to achieving the best possible accuracy and aligning with previous literature?

---

### Official Review · Reviewer_WsGs · 2024-11-04

**Soundness:** 3
**Presentation:** 3
**Contribution:** 2
**Rating:** 6
**Confidence:** 4

**Summary:**

This paper presents a framework for bias evaluation of unconditional image generative models. The authors measured the bias shift in the original and synthetic data and tested their framework in publicly available datasets. They found that, bias shift happens in image generative models and proposed two taxonomies to categorize the bias shift for different attributes. The paper is well-written and well-formulated. However, a comparison with the existing bias evaluation framework needs to be made.

**Strengths:**

1. This work presents a bias evaluation framework for unconditional image generative models.
2. The authors proposed two taxonomies for categorizing bias shifts for different attributes.
3. The authors experimented with different sizes of diffusion models to observe how bias shift is happening.

**Weaknesses:**

1. As this paper presents a bias evaluation framework for image dataset, it needs to be compared with other evaluation framework, i.e. compare with [1]. How is the presented framework differ with the [1]?

2. Limitations of this evaluation framework should be discussed in the paper.

#### References:

[1] Wang, Angelina, et al. "REVISE: A tool for measuring and mitigating bias in visual datasets." _International Journal of Computer Vision_ 130.7 (2022): 1790-1810.

**Questions:**

See weakness point 1

---

### Official Review · Reviewer_FJJr · 2024-11-04

**Soundness:** 2
**Presentation:** 3
**Contribution:** 2
**Rating:** 3
**Confidence:** 3

**Summary:**

This paper investigates how inductive bias in unconditional generative models affects bias in generated results. The authors define bias shift as the difference between the probability of attribute presence in the training and generated distributions, and train a classifier to categorize attributes to quantify bias shift. Furthermore, attributes are categorized as subjective or non-subjective based on the position of the classifier's decision boundary. The author validates multiple models including diffusion models and GAN on two datasets, CelebA and DeepFusion, revealing related patterns.

**Strengths:**

1. Problem Definition: The paper focuses specifically on studying the inductive bias of generative models themselves, avoiding other factors such as dataset bias and prompts, providing a novel perspective for analyzing bias sources.

2. Methodology: Proposes a standardized bias evaluation framework that uses the same classifier for all label predictions, ensuring consistency in evaluation.

3. Writing: The paper is well-structured and explains complex concepts in an understandable way.

**Weaknesses:**

1. Bias Definition: The paper's definition of bias, which only measures differences in attribute occurrence probabilities, may be overly simplistic. Bias typically encompasses more complex dimensions including social prejudices and systemic discrimination.
Oversimplified Metrics: The Average Bias Shift (ABS) metric may be too reductive as it:

2. ABS doesn't consider correlations between attributes and ignores the varying social impact weights of different attributes

3. The 0.01 threshold for subjective/non-subjective classification lacks justification, and there are no ablation studies on threshold selection. The data-driven categorization approach may overlook inherent social and ethical implications of attributes

4. Relying on a single classifier may introduce classifier-specific biases. Figures 4 and 5 show relatively low accuracy (90% or below) for many attributes, questioning the reliability of the pre-trained classifier. It would be better to consider using Large Language Models as supplementary evaluators (Just a suggestion, no need to add experiments).

5. The current model selection appears dated, primarily relying on ADM (2021) and BigGAN (2019) for experiments. They may not reflect the latest advances in generative modeling. The paper would benefit significantly from validating the proposed framework on more recent architectures, such as Stable Diffusion, DiT, PixArt-α for diffusion models, and StyleGAN3 for GANs.

**Questions:**

Do the accuracy rates reported in Figures 4 and 5 of the appendix refer to training set or validation set performance?

---

### Meta-Review · Area_Chair_RNWo · 2024-12-25

**Metareview:**

This work investigates the manifestation of bias in unconditional image generation process. The core idea is to use a classifier (trained on the same data that was used to train the generation process) as a tool to measure the bias gap between train and gen distributions.

The problem of studying inductive biases of generative models is an important one and an effort along this line is well appreciated.

There are several concerns raised by the reviewers regarding this work. The major ones are:

- the definition of bias which seems to be too simplistic leading to a metric called ABS which might not be effective enough to quantify biases.
- lack of proper justification behind the use of a single classifier as a tool to quantify biases -- a common concern by all the reviewers and myself. Several valid points have been raised regarding this and the rebuttal lacked convincing replies.
- outdated models under consideration (e.g., BigGAN)

This paper requires solid justifications behind the use of the classifier and better experiments on recent generative models.

**Additional Comments On Reviewer Discussion:**

- Reviewers raised several concerns regarding such as (1) the definition of bias term; (2) use of single classifier; (3) use of outdated models etc.
- Authors did engage well during the rebuttal however, the above concerns were not answered satisfactorily and unfortunately there was no clear indication towards the acceptance of this work in its current form.

---

### Decision · Program_Chairs · 2025-01-22

Reject